# GS-Blur: A 3D Scene-Based Dataset for Realistic Image Deblurring

**Dongwoo Lee**[1]    **Joonkyu Park**[1]    **Kyoung Mu Lee**[1,2]
[1]Dept. of ECE&ASRI, [2]IPAI, Seoul National University, Korea
{dongwoo.lee, jkpark0825, kyoungmu}@snu.ac.kr

## Abstract

To train a deblurring network, an appropriate dataset with paired blurry and sharp images is essential. Existing datasets collect blurry images either synthetically by aggregating consecutive sharp frames or using sophisticated camera systems to capture real blur. However, these methods offer limited diversity in blur types (blur trajectories) or require extensive human effort to reconstruct large-scale datasets, failing to fully reflect real-world blur scenarios. To address this, we propose GS-Blur, a dataset of synthesized realistic blurry images created using a novel approach. To this end, we first reconstruct 3D scenes from multi-view images using 3D Gaussian Splatting (3DGS), then render blurry images by moving the camera view along the randomly generated motion trajectories. By adopting various camera trajectories in reconstructing our GS-Blur, our dataset contains realistic and diverse types of blur, offering a large-scale dataset that generalizes well to real-world blur. Using GS-Blur with various deblurring methods, we demonstrate its ability to generalize effectively compared to previous synthetic or real blur datasets, showing significant improvements in deblurring performance. The dataset is available at: https://github.com/dongwoohhh/GS-Blur

## 1   Introduction

Single-image deblurring is a crucial challenge in image restoration, focusing on removing blur caused by motion between the camera and objects. To address this, pioneering approaches [23, 11, 37, 3] have proposed paired datasets, consisting of blurry images and their corresponding sharp images, designed for training deep neural networks. Specifically, their efforts to create deblurring datasets have primarily relied on two methods: *synthetic* [23, 22, 46] and *real* [47, 30, 29] data generation. However, both approaches heavily depend on heuristic human capture techniques, often leading to limitations such as incomplete coverage of large-scale datasets and inadequate representation of diverse blur types (*i.e.*, blur length and directions).

Since capturing blurry and sharp images simultaneously with a single sensor is challenging, earlier methods [35, 22, 33, 14] have resorted to synthetically generating blurry images from consecutive sharp frames. They achieve this by capturing consecutive sharp frames using high-speed cameras and then aggregating these neighboring frames to create synthetic blurry images. Although this allows for easy generation of blurry images, the resulting blur is derived from highly discrete frames, which leads to differences from real-world blur and fails to generalize well to real-world blurry images.

Later methods [47, 30, 29] have introduced specialized camera systems equipped with beam-splitters. These systems divide the light entering the camera lens into two image sensors with varying exposure times, producing sharp images from the shorter and blurry images from the longer-exposed sensor. While they generate a more realistic blur, making it better suited for real-world applications, they present several challenges. First, they require precise camera system design, complicating the use of diverse camera models. Indeed, [47] and [30] obtained their datasets from a single camera model, the

38th Conference on Neural Information Processing Systems (NeurIPS 2024) Track on Datasets and Benchmarks.

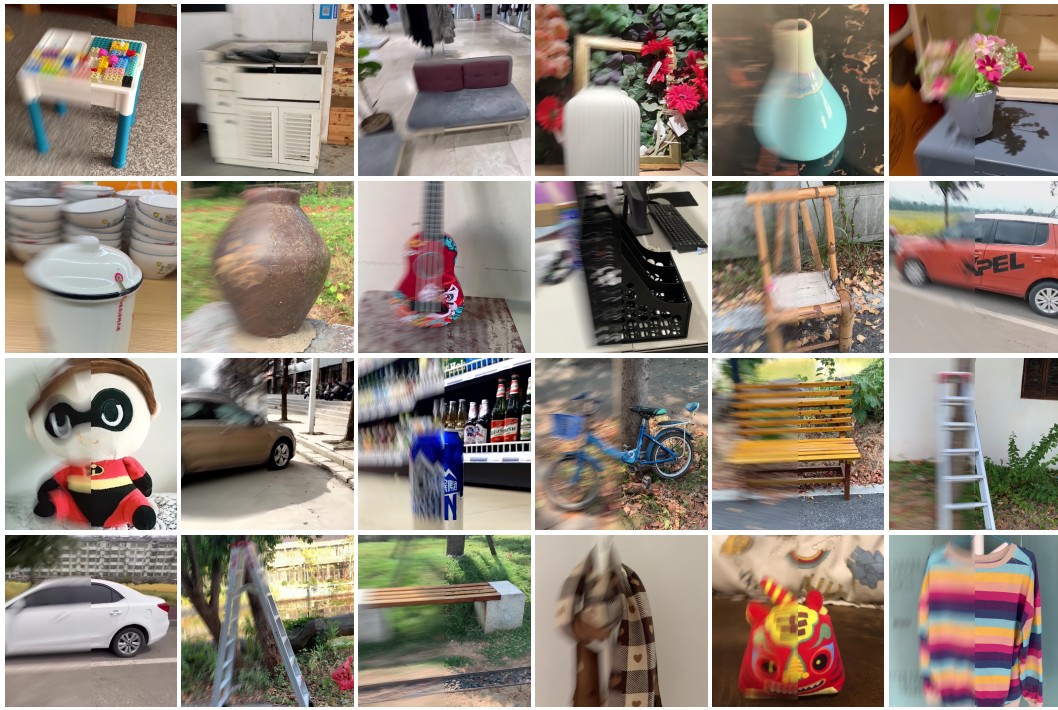

Figure 1: **Examples of the proposed GS-Blur dataset.** The left half of the frames displays synthetically generated blur, while the right half exhibits sharp pairs.

Sony A7R3 and machine vision camera, respectively. Second, despite the use of different exposure times for the two sensors, one capturing blurry images and the other sharp images, discrepancies in image signal processing (ISP) can arise, requiring additional image processing such as white balancing and color mapping. Moreover, the datasets heavily rely on human capture, which limits their scale and results in restricted blur trajectories in terms of blur length and direction.

In this paper, we explore methods to synthesize realistic blurry images to improve deblurring quality for real-world blurry images generally. To this end, we present a novel dataset, GS-Blur, which synthesizes blurry images using 3D Gaussian Splatting (3DGS) [9]. Specifically, we utilize the existing large-scale multi-view dataset, MVImgNet [43], to train 3DGS on sharp multi-view images, enabling the reconstruction of 3D scenes. Then, from these reconstructed scenes, we use two camera views to render images: one from a fixed position and one from a moving position along randomly generated motion trajectories, corresponding to sharp and blurry images, respectively. Specifically, following the method in [23, 22, 46], we aggregate multiple images from cameras along the trajectory to create blurry images, but unlike [23, 22, 46]'s use of highly discrete frames, we employ denser frames by positioning multiple cameras along the trajectories, resulting in more realistic blur. Moreover, by utilizing various degrees for the blur trajectories, our GS-Blur dataset includes diverse blur trajectories in terms of both blur length and direction. By using MVImgNet, which consists of large-scale multi-view images from diverse camera models, our GS-Blur provides diverse deblurring image pairs with significant advantages, showing generalizability, as detailed in the experiments Section 4.3. Additionally, we conduct comprehensive ablation studies to justify the reconstruction of our GS-Blur dataset in Section 4.4. The samples of GS-Blur are shown in Figure 1.

## 2   Related Works

**Image deblurring methods.** As blur commonly occurs in various situations [31, 24, 25, 28], early deblurring methods [11, 23, 26, 27, 3, 45, 15] modeled a blurry image as a convolution of a 2D blur kernel with a latent sharp image, optimizing the sharp image for a known blur kernel using Richardson-Lucy deconvolution. However, these methods struggled with real-world scenarios where

Table 1: **Comparison of existing datasets with our new GS-Blur dataset.**

| Dataset | Method | Scale | Exp. time (ms) | Resolution | Need ISP |
|---------|--------|-------|----------------|------------|----------|
| DVD [35] | Synthetic | 6,708 | 25 | 1280 × 720 | ✗ |
| GoPro [23] | Synthetic | 3,214 | 25-50 | 1280 × 720 | ✗ |
| REDS [22] | Synthetic | 30,000 | 25-50 | 1280 × 720 | ✗ |
| HIDE [33] | Synthetic | 8,422 | 41.667 | 1280 × 720 | ✗ |
| HFR-DVD [14] | Synthetic | 13,500 | 40 | 960 × 540 | ✗ |
| BSD [47] | Real | 33,000 | 1-24 | 640 × 480 | ✓ |
| RealBlur [30] | Real | 4,738 | 500 | 680 × 772 | ✓ |
| RSBlur [29] | Real+Synthetic | 13,358 | 100 | 1920 × 1200 | ✗ |
| **GS-Blur** | Synthetic | 156,209 | Various | Various | ✗ |

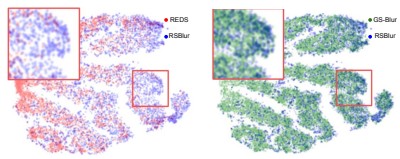

(a) Real vs Synthetic (b) Real vs **GS-Blur**

Figure 2: **Motion distribution visualization of synthetic, real, and GS-Blur datasets.**

blur kernels are unknown. Later, deep learning shifted the field to learning-based approaches, with models like DeblurGAN [11] and DeepDeblur [23] restoring sharp images without blur kernel estimation. Advanced models such as NAFNet [3] and Restormer [45] use channel attention modules, while SwinIR [15] introduce Vision Transformer [5, 18] architectures. Despite architectural advances, these methods are data-driven, requiring extensive training data that aligns well with real-world blur for effective generalization.

**Deblurring datasets.** Table 1 shows the overview of deblurring datasets. Earlier, several synthetic datasets [35, 23, 22, 33, 14, 46] using high-speed cameras have been proposed to train learning-based deblurring methods. DVD [35], GoPro [23], and REDS [22] create blurry images by averaging consecutive sharp frames to simulate motion blur. Similarly, HIDE [33] synthesizes blur with densely annotated foreground human bounding boxes, and HFR-DVD[14] uses a higher-speed camera (*e.g.*, SONY DSC-RX10 IV) to capture video frames, resulting in more realistic blurs. However, even with high-speed cameras, the time interval between frames is too discrete to accurately capture continuous real-world blur patterns, resulting in less effective generalization to real-world blurry images.

Other approaches [47, 30, 29] use beam splitter camera systems to capture paired images, addressing synthetic dataset limitations. These systems capture blurry images with longer exposure and sharp images with shorter exposure, accurately mimicking real-world blur. However, they face several challenges. First, they require the precise design of the camera system, which is a labor-intensive task. Moreover, due to the need for a sophisticated system, these datasets are restricted to specific camera models, showing less generalizability to blurry images captured by different cameras. Despite the implementation of the specialized camera system, utilizing different exposure times for the two image sensors, where one captures blurry images and the other sharp images, can lead to discrepancies in different light intensities reaching each sensor. This difference requires processing the images with different ISO settings, leading to disparities in tone or color between the blurry and sharp images. As a result, additional ISP processing is required to match them. Furthermore, their motion trajectories highly rely on humans, failing to capture the diverse blur patterns found in real-world scenarios.

In contrast, our GS-Blur, though synthetically derived, offers greater scale, diverse exposure times, and various resolutions compared to previous datasets, as shown in Table 1. Furthermore, the motion trajectories in GS-Blur are randomly generated in 3D space, effectively encompassing potential real-world motion trajectories. Figure 2 compares the motion trajectory distributions of synthetic [22] and real deblurring datasets [29] with our GS-Blur dataset. Here, each point in the figure represents the t-SNE [39] projection of a motion trajectory, computed using the optical flow [38] from the provided consecutive sharp frames. As shown in Figure 2a, the distribution of the synthetic dataset (●) does not overlap with the distribution of the real dataset (●), indicating that the previous synthetic dataset fails to cover the motion diversity of real-world images. On the other hand, Figure 2b shows that the distribution of GS-Blur (●) overlaps with the real distribution, demonstrating that GS-Blur covers most real blurry images' distributions and exhibits a wider range of blur diversity.

**Novel view synthesis.** Unlike previous synthetic datasets that simulate blur by aggregating images, our approach recovers 3D scenes and moves the view within these 3D spaces to mimic camera shakes with varying trajectories, making our work closely related to novel view synthesis. Earlier, Neural Radiance Fields (NeRF) [20] made significant strides in 3D vision tasks, particularly in photo-realistic novel view synthesis. Despite the potential of NeRF's implicit neural representation (INR) to become a widely used 3D representation, NeRF-based methods [36, 6, 1, 21] face challenges in achieving real-time novel view synthesis without compromising visual quality. To address this, 3D Gaussian Splatting (3DGS) [9] streamlines the NeRF framework through point-based 3D Gaussians and tile-based rasterization, enabling high-quality, real-time novel view synthesis at 1080p resolution.

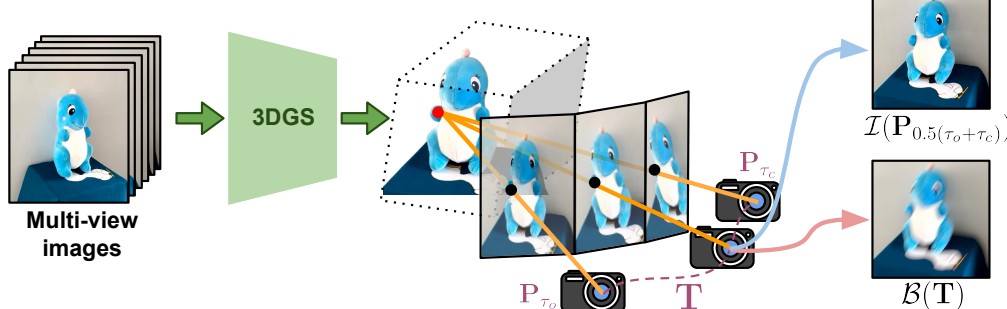

Figure 3: **The overall pipeline for generating blurry and sharp image pairs in our GS-Blur dataset.** To this end, we first train a 3D Gaussian Splatting model to reconstruct 3D scenes from multi-view images. Then, from these reconstructed 3D scenes and randomly generated motion trajectory $\mathbf{T}$, we render sharp images $\mathcal{I}(\mathbf{P}_{0.5(\tau_o+\tau_c)})$ from a fixed camera view and blurry images $\mathcal{B}(\mathbf{T})$ from a moving camera view. Specifically, we render $\mathcal{M}$ sharp images along the motion trajectory and then average these sharp frames to synthesize the blurry image.

## 3 GS-Blur dataset

### 3.1 Preliminary: 3D Gaussian Splatting

3D Gaussian Splatting (3DGS) [9] models a 3D scene from multi-view images using Gaussian primitives $\{\boldsymbol{\mu}_k, \boldsymbol{\Sigma}_k, \sigma_k, \boldsymbol{S}_k\}_{k\in\mathcal{K}}$, where each parameter represents the position $\boldsymbol{\mu}_k$, covariance $\boldsymbol{\Sigma}_k$, opacity $\sigma_k$, and spherical harmonic coefficients $\boldsymbol{S}_k$ of a sparse 3D point $k \in \mathcal{K}$, initialized from SfM [32]. When rendering an image, the Gaussian primitives are projected onto the camera's image plane, and the color of each pixel $\boldsymbol{p}$ is computed using point-based $\alpha$-blending [48] as follows:

$$\hat{C}(\boldsymbol{p}) = \sum_{k\in\mathcal{K}} \alpha_k \mathbf{c}(\mathbf{v}_k; \boldsymbol{S}_k) \prod_{j=1}^{k-1}(1 - \alpha_j),\tag{1}$$

$$\text{where} \quad \alpha_k = \sigma_k e^{-\frac{1}{2}(\boldsymbol{p}-\boldsymbol{\mu}_k^\downarrow)^T \boldsymbol{\Sigma}_k^{\downarrow-1}(\boldsymbol{p}-\boldsymbol{\mu}_k^\downarrow)}.\tag{2}$$

The color of the $k$-th Gaussian is computed using the spherical harmonic function $\mathbf{c}(\mathbf{v}_k; \boldsymbol{S}_k)$ for the camera's viewing direction $\mathbf{v}_k$, and the density $\alpha_k$ is determined from the 2D projected Gaussian weights $\boldsymbol{\mu}_k^\downarrow$ and $\boldsymbol{\Sigma}_k^{\downarrow-1}$ as introduced in [48]. The point-based $\alpha$-blending in Equation 1 essentially follows the same image formation model as NeRF [20]. However, compared to NeRF, the explicit representation of Gaussian primitives allows for significantly faster rendering. Specifically, the cost-effective Gaussian rasterization replaces the computationally intensive multi-layer perception and ray-point sampling approach used in NeRF. As a result, 3DGS achieves high-quality real-time view synthesis and reduces training time to tens of minutes. In this work, we use the fast training and rendering speeds of 3DGS to generate realistic blurry images by densely sampling views along a moving camera trajectory.

### 3.2 Preliminary: MVImgNet dataset

MVImgNet [43] is a large-scale multi-view image dataset comprising 6.5 million frames from 219,199 videos, covering objects from 238 classes. These videos are captured using various common cameras (*e.g.*, smartphones), reflecting a diverse range of real-world image distributions. On this basis, we leverage this dataset to reconstruct our GS-Blur dataset. Specifically, we manually selected 26 classes suitable for constructing a deblurring dataset, with the detailed class information provided in our appendix.

### 3.3 Pipeline for blur synthesis of GS-Blur

Figure 3 provides an overview of the construction process for our GS-Blur dataset. To collect GS-Blur, we first train 3DGS using a set of $\mathcal{N}$ posed sharp images $\{\mathcal{I}(\mathbf{P}_i)\}_{i\in\mathcal{N}}$ with their corresponding

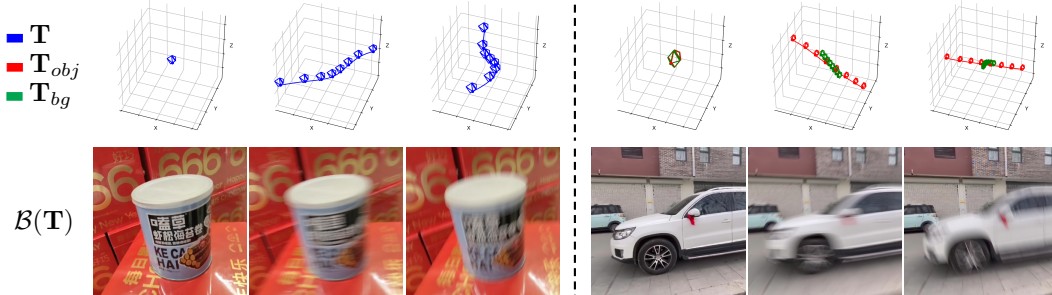

Figure 4: **Visualization of randomly generated 3D trajectories and their corresponding motion-blurred images** $\mathcal{B}(\mathbf{T})$. **(Left)** By using different trajectories $\mathbf{T}$ for different images, we can generate multiple blurry images corresponding to a single sharp image. Here, we use the same trajectory $\mathbf{T}$ for both the object and the background. **(Right)** By using different motion trajectories for the object and background, $\mathbf{T}_{obj}$ and $\mathbf{T}_{bg}$, respectively, we can simulate rigid-body motion blur. Note that the first and fourth columns in the figure show sharp images with fixed camera views.

camera poses $\{\mathbf{P}_i\}_{i \in \mathcal{N}}$. Here, each scene is trained with the 3DGS model for 30,000 iterations. Then, from the reconstructed 3D scenes, we render an image $\mathcal{I}(\mathbf{P})$ from any arbitrary camera view $\mathbf{P}$. Specifically, to simulate the process of capturing blurry images in real-world scenarios, where moving cameras create blurred images while the shutter is open, we mimic this by moving the camera along a 3D trajectory. We then create the blurry image by accumulating multiple rendered images, each captured by a camera along the motion trajectory. Let $\mathbf{T} = \{\mathbf{P}_\tau\}_{\tau \in [\tau_o, \tau_c]}$ denote continuous camera poses along the trajectory that generates a motion-blurred image $\mathcal{B}(\mathbf{T})$, and we can synthesize the blurred image from rendered sharp images of 3DGS as follow:

$$\mathcal{B}(\mathbf{T}) = g\left(\int_{\tau_o}^{\tau_c} g^{-1}(\mathcal{I}(\mathbf{P}_\tau))d\tau\right) \simeq g\left(\frac{1}{\mathcal{M}}\sum_{t=1}^{\mathcal{M}} g^{-1}(\mathcal{I}(\mathbf{P}_t))\right). \quad (3)$$

Here, the Camera Response Function (CRF) $g$ maps an image from linear RGB space to sRGB space, with $g^{-1}$ being its inverse function. We accumulate rendered sharp images in the linear space and then convert the accumulated blurry image to sRGB space, following the approach in [23, 22]. In real cameras, the RGB color is continuously accumulated while the shutter is open. To simulate this continuous accumulation, we approximate it using a finite sum of $\mathcal{M}$ intermediate sub-frames, which is valid when $\mathcal{M}$ is sufficiently large. Different from the real camera systems, sub-frame rendering through 3DGS does not degrade image quality, regardless of the $\mathcal{M}$ value. In practice, we set $\mathcal{M} = 121$ and select the middle sub-frame as the ground truth sharp image pair. In other words, the sharp images are rendered from a fixed camera position at $\mathbf{P}_{0.5(\tau_o+\tau_c)}$; therefore, a sharp image is represented as $\mathcal{I}(\mathbf{P}_{0.5(\tau_o+\tau_c)})$.

**Random Blur Generation.** Generating deblur data through novel view synthesis offers the distinct advantage that blurry images can be synthesized from randomly generated camera motions. While any kind of polynomial curve or spline model can function as a camera motion trajectory, we adopt the random-order Bézier curve, which is widely used in prior works [34, 13, 12].

For the camera motion generation given initial camera pose $\mathbf{P}_i$ in training views, we follow the subsequent procedures: 1) Randomly generate a linear motion trajectory in 6 degrees of freedom (6DOF) pose space. 2) Generate an $n^{th}$-order Bézier curve by randomly perturbing the points that divide the linear motion trajectory into $n + 1$ equal segments. 3) Align the center pose of Bézier curve to be $\mathbf{P}_i$ and sample $\mathcal{M}$ camera poses from the curve. Note that we randomly select the curve parameters $\{n, \delta_t, \delta_r\}$, where $n \in [1, 5]$ denotes the order of Bézier curve, $\delta_t \in \mathbb{R}^3$ represents the length of the curve and $\delta_r \in \mathbb{R}^3$ indicates the shift in orientation, respectively. Here, the 3 dimensions of $\delta_t$ and $\delta_r$ correspond to the $x$-, $y$-, and $z$-axis of the 3D space.

Since the randomly selected curves, which correspond to camera movements, directly affect the formation of blur, we choose the parameters $\delta_t$ and $\delta_r$ within pre-defined boundaries to reflect realistic blur. Specifically, we randomly sample the 3D length $\delta_t$ from the range $[0, 0.7]$, considering the blur

length in previous datasets, and the 3D orientation $\delta_r$ from the range [-1.5°, 1.5°], accounting for the minimal impact of rotation during short exposure times in real-world blurry image capturing.

**1-to-n blur generation.** An additional advantage of the proposed dataset generation is the capability for 1-to-n blur generation. Existing datasets [23, 22, 47, 29] collected with high-speed or beam-splitter cameras typically yield only one blurred image per sharp image or adjust the blur magnitude by altering the number of frames synthesized. In contrast, our method allows for the synthesis of multiple corresponding blurry images for a single sharp image by generating independent trajectories multiple times, which is crucial in preventing overfitting in deblurring architectures. Figure 4 (left) displays examples of multiple (n) blurry images $\mathcal{B}(\mathbf{T})$ corresponding to a single sharp image, using different blur trajectories.

**Rigid-body Object Motion Blur.** The primary limitation of generating deblur data with 3DGS is its restriction to rendering static scenes, allowing only for motion blur caused by camera movement. However, in real-world blurry images, motion blur often arises from moving objects like pedestrians or vehicles, independent of camera motion. To address this, we leverage the object's binary segmentation mask $\mathbf{m}_s \in \{0, 1\}$ to simulate rigid-body motion. Specifically, we generate two random motion trajectories: one trajectory $\mathbf{T}_{obj}$ to create rigid-body motion blur for the object $\mathcal{B}(\mathbf{T}_{obj})$, and another trajectory $\mathbf{T}_{bg}$ to simulate camera motion blur in the background $\mathcal{B}(\mathbf{T}_{bg})$. Using these two trajectories and the object mask $\mathbf{m}_s$, we apply alpha matting to produce a blurry image where the object and background are distinctly blurred by their respective motions. Here, the alpha value $\mathbf{m}_s(\mathbf{T}_{obj})$ for the mapping is calculated by averaging after 3D warping $\mathbf{m}_s$ along $\mathbf{T}_{obj}$ as follows:

$$\mathbf{m}_s(\mathbf{T}_{obj}) = \sqrt{\frac{1}{\mathcal{M}} \sum_{t=1}^{\mathcal{M}} \pi(\mathbf{m}_s; \mathbf{P}_t)}, \tag{4}$$

$$\mathcal{B}(\mathbf{T}_{obj}, \mathbf{T}_{bg}) = \mathbf{m}_s(\mathbf{T}_{obj}) \cdot \mathcal{B}(\mathbf{T}_{obj}) + (1 - \mathbf{m}_s(\mathbf{T}_{obj})) \cdot \mathcal{B}(\mathbf{T}_{bg}). \tag{5}$$

The object mask of each sub-frame is computed by backward warping [7] $\pi(\mathbf{m}_s; \mathbf{P}_t)$, where the camera intrinsic and the depth and the pose of the sub-frame $t$ are parameters of the warping function $\pi :\in \mathbb{R}^{H \times W} \mapsto \mathbb{R}^{H \times W}$. Note that applying the square-root to the alpha value results in more natural blending at object boundaries, since the background color has already been mixed at the boundaries when synthesizing $\mathcal{B}(\mathbf{T}_{obj})$. Figure 4 (right) shows examples of blurry images $\mathcal{B}(\mathbf{T}_{obj}, \mathbf{T}_{bg})$ generated using different random motion trajectories, $\mathbf{T}_{obj}$ and $\mathbf{T}_{bg}$, for object and background, respectively.

**Noise addition.** 3DGS employs spherical harmonics to model view-dependent RGB colors, which leads to smooth renderings even when the input images contain slight noise. However, using these smooth renderings to train deblurring deep networks diminishes their generalizability to real-world blurry images, since the networks may fail to learn the necessary features that are typical in naturally occurring noise and complex blur variations. Therefore, we integrate the realistic blur synthesis pipeline introduced in RSBlur [29] to synthesize realistic image noise into the blurred renderings generated from 3DGS. To this end, we convert images from the sRGB space to the camera RAW space, then add Poisson and Gaussian noises, and finally convert them back to the sRGB space, approximating the noise generation principles that occur in real camera systems.

**Multi-Resolution.** As MVImgNet [43], the source of our GS-Blur, predominantly contains object-centric scenes where objects are often captured close to the camera view, considerable pixels of the rendered images may consist solely of objects. However, during training deblurring network, image patches are typically cropped to smaller size (*e.g.*, 256×256), potentially leading to ineffective training due to the overwhelming presence of objects. To address this, we introduce random down-scaled renderings $\{\times 1/2, \times 1/3, \times 1/4\}$ from the rendered high-resolution images (*e.g.*, 1920×1080) as data augmentation, enabling a broader 3D region to be included within the cropped image patch. However, note that our downsampling differs from that of previous datasets [14, 22], where downsampling is aimed at reducing noise. In our case, we add noise after downsampling.

Finally, we reconstruct $3,408$ scenes from the subset of MVImgNet and train 3DGS to obtain $156,209$ sharp renderings for blur generation. By rendering multiple random blurry pairs and utilizing down-scaled rendering augmentation, we have constructed a GS-Blur dataset consisting of a total of $752,335$ blurry images.

Table 2: **Quantitative comparison of cross-validation regarding PSNR and SSIM.** We train various networks [4, 40, 3] on different training sets and assess their performance in various testing sets. The highest-performing models are indicated in blue, while the second-best performers are highlighted in red. The GS-Blur-trained model demonstrates the best performance across most scenarios, except when the training and testing set exactly match.

| Train Set | Test Set Metrics | GoPro | | | REDS | | | BSD | | | RSBlur | | |
|---|---|---|---|---|---|---|---|---|---|---|---|---|---|
| | | MIMO | UFormer | NAFNet | MIMO | UFormer | NAFNet | MIMO | UFormer | NAFNet | MIMO | UFormer | NAFNet |
| GoPro | PSNR | 31.21 | 32.32 | 32.81 | 27.12 | 29.03 | 27.26 | 22.44 | 27.78 | 26.25 | 21.75 | 27.53 | 24.97 |
| | SSIM | 0.915 | 0.934 | 0.960 | 0.817 | 0.868 | 0.872 | 0.708 | 0.861 | 0.831 | 0.539 | 0.720 | 0.727 |
| REDS | PSNR | 26.75 | 27.11 | 27.67 | 32.99 | 32.74 | 33.49 | 27.35 | 26.55 | 28.08 | 28.72 | 27.31 | 24.93 |
| | SSIM | 0.822 | 0.848 | 0.903 | 0.921 | 0.931 | 0.946 | 0.833 | 0.837 | 0.873 | 0.717 | 0.716 | 0.739 |
| BSD | PSNR | 27.13 | 28.17 | 28.38 | 27.90 | 28.23 | 28.23 | 33.43 | 33.30 | 33.87 | 29.89 | 30.49 | 30.63 |
| | SSIM | 0.831 | 0.859 | 0.909 | 0.838 | 0.854 | 0.886 | 0.940 | 0.942 | 0.953 | 0.781 | 0.799 | 0.843 |
| RSBlur | PSNR | 28.50 | 28.62 | 30.24 | 27.67 | 27.79 | 28.60 | 29.72 | 30.58 | 31.43 | 32.52 | 32.97 | 33.72 |
| | SSIM | 0.868 | 0.895 | 0.937 | 0.830 | 0.851 | 0.891 | 0.897 | 0.918 | 0.931 | 0.836 | 0.851 | 0.891 |
| **GS-Blur** | PSNR | 29.44 | 30.80 | 31.49 | 29.55 | 30.19 | 30.54 | 30.42 | 31.05 | 31.37 | 31.17 | 31.86 | 32.30 |
| | SSIM | 0.882 | 0.914 | 0.949 | 0.876 | 0.894 | 0.924 | 0.910 | 0.924 | 0.941 | 0.812 | 0.836 | 0.868 |

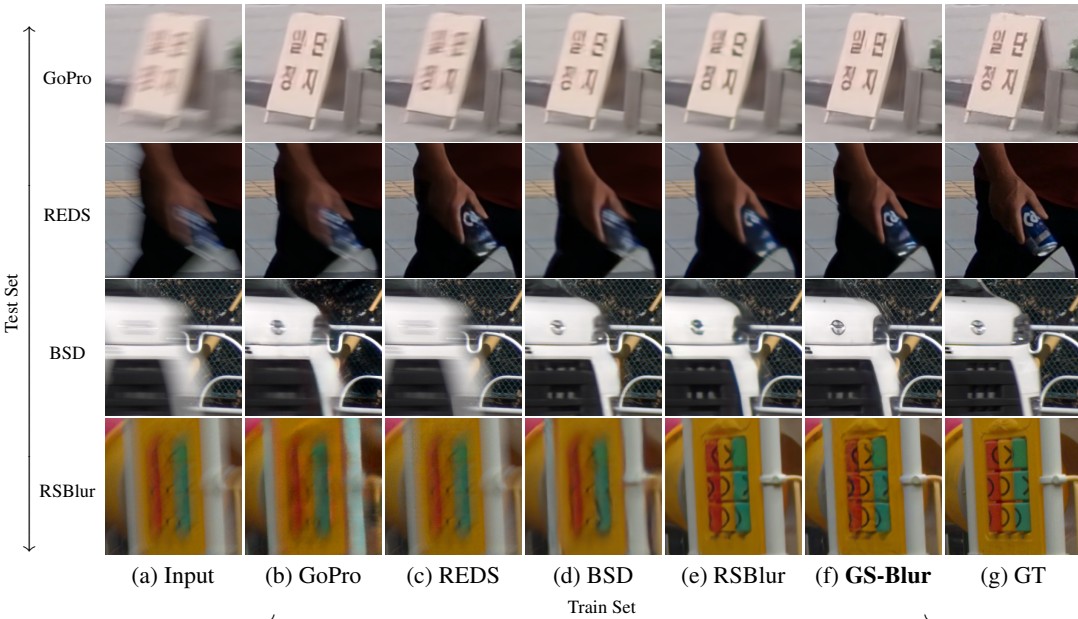

(a) Input    (b) GoPro    (c) REDS    (d) BSD    (e) RSBlur    (f) **GS-Blur**    (g) GT

Train Set

Figure 5: **Qualitative comparison of cross-validation.** We present visual comparisons using NAFNet [3] trained on different datasets, as indicated in the caption. Except when the training and test sets match, the model trained on our GS-Blur (f) consistently produces the most visually appealing results.

## 4 Experiments

### 4.1 Implementation details

To evaluate the efficacy of our GS-Blur dataset, we employ recent state-of-the-art deblurring architectures, including Transformer-based architectures (Uformer [40]) and a CNN-based architecture (MIMO-UNet [4] and NAFNet [3]), following their respective training protocols. Specifically, the deblurring networks are trained on random crops of size $256 \times 256$, utilizing a batch size of 4 for MIMO-UNet and Uformer and 8 for NAFNet per GPU, with 4 NVIDIA Quadro RTX 8000. Random horizontal and image rotations are also applied to training samples according to each network's protocol, totaling 200k iterations. For Uformer, cosine annealing [19] is employed, starting from $2e^{-4}$ and decaying to $1e^{-6}$, while for NAFNet, it starts from $1e^{-3}$ and decays to $1e^{-6}$. In the case of MIMO-UNet, the learning rate is halved every 30k iterations. For the metrics, we evaluate the results using conventional image quality assessment metrics such as PSNR and SSIM [41].

Table 3: **Quantitative comparison on real blurry images [35].**

| Train Set | MUSIQ [16] | TOPIQ [2] |
|---|---|---|
| GoPro | 37.305 | 0.262 |
| REDS | 38.059 | 0.261 |
| BSD | 45.192 | 0.318 |
| RSBlur | 41.058 | 0.314 |
| **GS-Blur** | **48.004** | **0.332** |

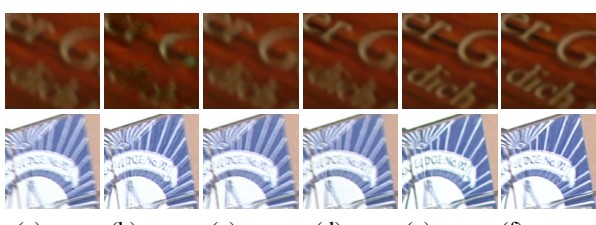

(a) Input  (b) GoPro  (c) REDS  (d) BSD  (e) RSBlur  (f) **GS-Blur**

Figure 6: **Visual comparison on real blurry images [35].** Captions represent the training dataset.

Table 4: **Quantitative comparison when training NAFNet [3] on various thresholding strategy.** In the table, 'Failed' refers to 3D scenes with significant drops in PSNR that did not meet the threshold, while 'Passed' refers to scenes with favorable PSNR that successfully passed the threshold.

| PSNR thresholding | # Scenes | GoPro PSNR | GoPro SSIM | REDS PSNR | REDS SSIM | BSD PSNR | BSD SSIM | RSBlur PSNR | RSBlur SSIM |
|---|---|---|---|---|---|---|---|---|---|
| Failed | 1622 | 31.01 | 0.945 | 30.26 | 0.919 | 31.36 | 0.934 | 32.05 | 0.865 |
| Passed + Failed | 5030 | 31.46 | 0.948 | 30.52 | 0.923 | **31.42** | 0.935 | 32.13 | 0.866 |
| Passed (Ours) | 3408 | **31.49** | **0.949** | **30.54** | **0.924** | 31.37 | **0.941** | **32.30** | **0.868** |

## 4.2 Reliability of sharp frames

Unlike previous datasets [23, 22, 30, 47, 29], where only blurry images are generated through a synthetic pipeline, both sharp and blurry images in our GS-Blur dataset are synthesized through rendering. As a result, the generated sharp images may contain floating point artifacts that do not accurately represent clean images. To address this, we have measured the PSNR between the ground truth and rendered sharp images for each 3D-reconstructed scene, removing scenes that fall below a certain PSNR threshold. Specifically, if any view showed a PSNR drop of more than 3dB from the mean, the entire scene was classified as failed and excluded from the dataset. This method ensures that only high-quality scenes are included, reducing the impact of floating point-induced blurring in the GS-Blur dataset. Finally, we evaluated our dataset by measuring PSNR and SSIM from multi-view images captured by camera angles not used in 3DGS training, resulting in PSNR=36.73 and SSIM=0.957, confirming the reliability of our sharp frames.

## 4.3 Generalization of blurry frames

**Cross-validation with previous deblurring datasets.** To demonstrate that our dataset generalizes well to diverse blurry images, Table 2 compares cross-validation results using our GS-Blur dataset with conventional deblurring datasets, including synthetic [23, 22] and real [47, 29] blurry images. The results indicate that, except when the training and evaluation sets match, models trained on our GS-Blur dataset consistently achieve the best results. This highlights our dataset's generalizability to both synthetically generated and real blurry images, regardless of the used model architectures. Furthermore, Figure 5 shows visual results from various benchmark datasets [23, 22, 47, 29] using NAFNet [3] trained on different datasets. As shown, except when the training and testing sets match, the model trained on our GS-Blur consistently delivers satisfactory results across all benchmark datasets. Please refer to the appendix for visual comparisons using other models [4, 40].

**Generalization on real blurry images.** While BSD and RSBlur provide realistic blurry images using a beam splitter, their specialized cameras (*e.g.*, machine vision) may differ from other camera models. Therefore, we present a quantitative comparison on real-blurry images [35] in Table 3. Since this dataset contains only blurry images without sharp counterparts, we use recent non-reference-based metrics (*e.g.*, MUSIQ [16] and TOPIQ [2]) for evaluation. As shown, the model trained on our GS-Blur achieves the best results, with visual comparisons available in Figure 6.

Table 5: **Deblurring performance comparison when training NAFNet [3] on GS-Blur with various blur generation pipelines.** The cross marker ✓and ✗indicate whether the corresponding component is applied or not for reconstructing the GS-Blur dataset, respectively. The last row represents our final GS-Blur dataset.

| 1-to-n blur | Rigid-body obj motion | Noise addition | Multi Resolution | Metrics | GoPro | REDS | BSD | RSBlur |
|---|---|---|---|---|---|---|---|---|
| ✗ | ✗ | ✗ | ✗ | PSNR | 30.69 | 30.40 | 29.69 | 30.18 |
|  |  |  |  | SSIM | 0.940 | 0.921 | 0.913 | 0.808 |
| ✗ | ✓ | ✓ | ✓ | PSNR | 31.39 | 30.19 | 31.13 | 32.23 |
|  |  |  |  | SSIM | 0.948 | 0.922 | 0.931 | 0.867 |
| ✓ | ✗ | ✓ | ✓ | PSNR | 31.44 | 30.27 | 31.11 | 32.29 |
|  |  |  |  | SSIM | 0.948 | 0.922 | 0.931 | 0.867 |
| ✓ | ✓ | ✗ | ✓ | PSNR | 30.97 | **30.94** | 30.36 | 30.99 |
|  |  |  |  | SSIM | 0.944 | **0.931** | 0.918 | 0.816 |
| ✓ | ✓ | ✓ | ✗ | PSNR | 31.24 | 30.07 | 31.06 | 32.23 |
|  |  |  |  | SSIM | 0.947 | 0.918 | 0.930 | 0.867 |
| ✓ | ✓ | ✓ | ✓ | PSNR | **31.49** | 30.54 | **31.37** | **32.30** |
|  |  |  |  | SSIM | **0.949** | 0.924 | **0.941** | **0.868** |

### 4.4 Ablation studies

We reconstruct blurry images in GS-Blur by excluding 3D scenes under certain PSNR thresholds, 1-to-n blur generation, rigid-body object motion blur, noise addition, and multiple resolutions. This section validates the effectiveness of each component by evaluating the deblurring network [3] trained on our GS-Blur with various modifications on previous benchmark datasets [23, 22, 47, 29]. The overall results are presented in Tables 4 and 5. Specifically, when comparing the last row of Table 4 with the other rows, our PSNR thresholding strategy clearly improves deblurring performance, demonstrating the effectiveness of the components used to reconstruct GS-Blur. Similarly, when comparing the first and last rows of Table 5, our strategy of using four components significantly improves deblurring performance, showing the efficacy of the used components to reconstruct GS-Blur. In the following sections, we illustrate the effectiveness of each component by individually removing them from our final GS-Blur.

**PSNR thresholding.** As described in Section 4.2, we construct our GS-Blur dataset by excluding 3D scenes with significant drops in PSNR. To demonstrate the effectiveness of this PSNR thresholding strategy, Table 4 compares results for GS-Blur with and without employing PSNR thresholding. When the deblurring network [3] is trained solely on the dataset from failed scenes (the first row of Table 4), there is a clear decline in deblurring performance, indicating that scenes with inaccurate 3D reconstruction (*e.g.*, floating point artifacts) hinder training. In contrast, training on our filtered dataset (the third row of Table 4), which excludes scenes that fall below the PSNR threshold, consistently outperforms the unfiltered dataset (the second row of Table 4), even with fewer scenes. This confirms the effectiveness of our filtering approach.

**1-to-n blur generation.** Unlike previous datasets [23, 22, 47, 29], which provide only single blurry image per sharp image and may lead to overfitting, GS-Blur can generate multiple (n) blurry images from a single sharp image by varying the motion trajectories (see Figure 4 (left)). As shown in the second row of Table 5, the performance drops consistently without 1-to-n blur generation, as our final approach better aligns with real-world scenarios where diverse blurs can originate from a single sharp image.

**Rigid-body object motion.** In real-world images, blur is caused not only by camera shake but also by moving objects, resulting in varying motion trajectories across different image pixels. To replicate this scenario, we incorporate rigid-body object motion into the GS-Blur dataset by applying distinct motion trajectories $\mathbf{T}_{obj}$ for objects and $\mathbf{T}_{bg}$ for the background, effectively simulating situations like a moving car captured from a stationary camera (see Figure 4 (right)). Compared to the third row

of Table 5, our approach in the last row, which incorporates rigid-body object motion, consistently enhances performance across various benchmark datasets.

**Noise addition.** Real-world images often contain noise, and adding noise to blurry images can make them more representative of real-world scenarios, leading to improved performance. Specifically, as both synthetic datasets (GoPro [23]) and real datasets (BSD [47] and RSBlur [29]) include noise in their testing sets, training the deblurring network on our noise-added GS-Blur dataset results in mostly better performance, as compared in the fourth and the last rows of Table 5. However, REDS [22] suppresses noise in its evaluation set by downscaling the images by $\frac{2}{3}$, leading to worse performance when the network is trained on the noise-added GS-Blur. Nevertheless, since blurry images from real datasets [47, 29] often contain noise, the addition of noise significantly enhances performance on those datasets.

**Multi-resolution.** When comparing the fifth and last rows of Table 5, which represent the performance with fixed- and multiple-resolution images in the GS-Blur dataset reconstruction, there is consistent improvement in overall performance with multiple-resolution images. This improvement can be attributed to the characteristics of the MVImgNet dataset [43], which mainly contains objects close to the camera. By incorporating multiple resolutions, our approach effectively simulates various distances between the camera and objects, resulting in more realistic images and consistently better performance across the evaluated metrics.

## 5 Limitations

While our GS-Blur dataset effectively mimics real blur and demonstrates its value through cross-validation across various benchmark datasets [23, 22, 47, 29], it has two potential limitations. First, although the GS-Blur dataset mimics real blur by moving the camera view along random blur trajectories and simulates rigid-body object motion with different blur trajectories for objects $\mathbf{T}_{obj}$ and background $\mathbf{T}_{bg}$, it cannot consider objects that change shape over time. For example, since the 3D scenes are based on static images, dynamic actions such as the movement of pedestrians' arms and legs during walking or the rotating wheels of a moving vehicle are not represented in GS-Blur. However, by utilizing recent advancements in 4D Gaussian Splatting [42, 8, 17], which can reconstruct temporal 3D scenes from multi-view video inputs, we plan to expand our dataset in future work to include such dynamic changes. Second, unlike conventional sharp images captured directly from cameras, our sharp images are rendered from 3D scenes, which may introduce a gap between them and real-world clean images. Nonetheless, we believe that recent advancements in 3D reconstruction and single-image generation [44, 10] could improve our method, leading to a more accurate dataset reconstruction of our GS-Blur dataset.

## 6 Conclusion

In this paper, we introduce the GS-Blur dataset, the first deblurring dataset reconstructed from 3D scenes. Unlike previous methods that struggle to obtain diverse blur trajectories, our approach easily simulates various blur trajectories by moving the camera view within the 3D scenes. By using our dataset, we demonstrate improved deblurring quality both qualitatively and quantitatively across various benchmark datasets and deblurring networks, demonstrating the high generalizability of GS-Blur. Furthermore, through extensive experiments, we validate the effectiveness of each component in the pipeline used to reconstruct blurry images in our GS-Blur dataset.

## Acknowledgments

This work was supported in part by the IITP grants [No.2021-0-01343, Artificial Intelligence Graduate School Program (Seoul National University), No. 2021-0-02068, and No.2023-0-00156], the NRF grant [No. 2021M3A9E4080782] funded by the Korea government (MSIT), and the SNU-LG AI Research Center.

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

# Appendices

## A    Additional visual comparison of cross-validation

Figures 7, 8, and 9 present a comprehensive qualitative comparison of cross-validation results using three different deblurring models: MIMO-UNet [4], UFormer [40], and NAFNet [3]. These extensive qualitative results highlight the effectiveness of our proposed GS-Blur dataset in enhancing the generalizability of deblurring for real-world motion blur.

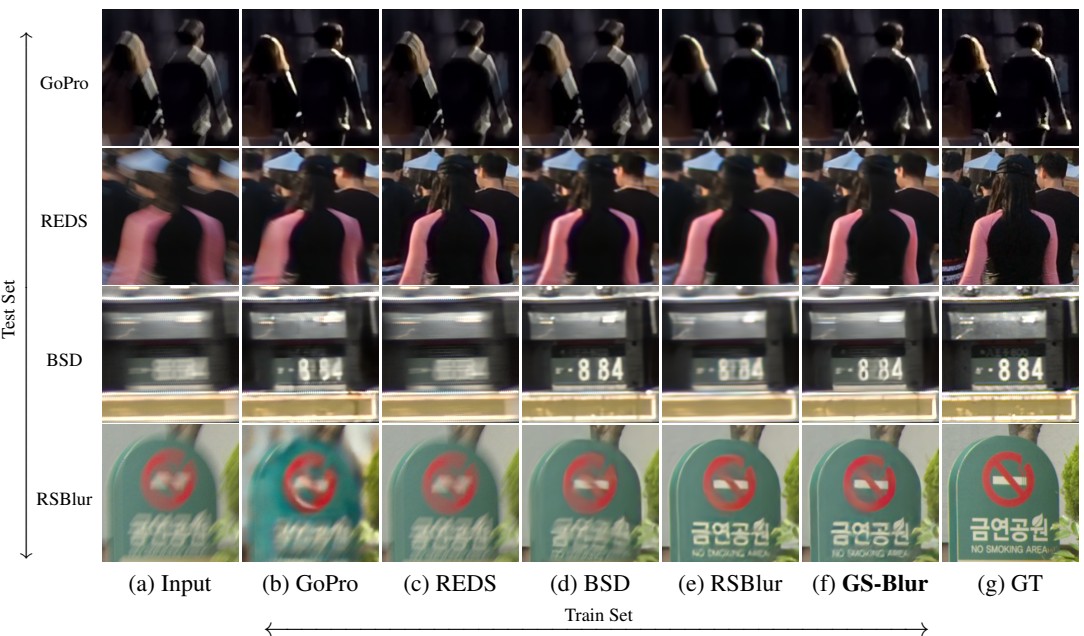

Figure 7: **Qualitative comparison of cross-validation using MIMO-UNet [4].**

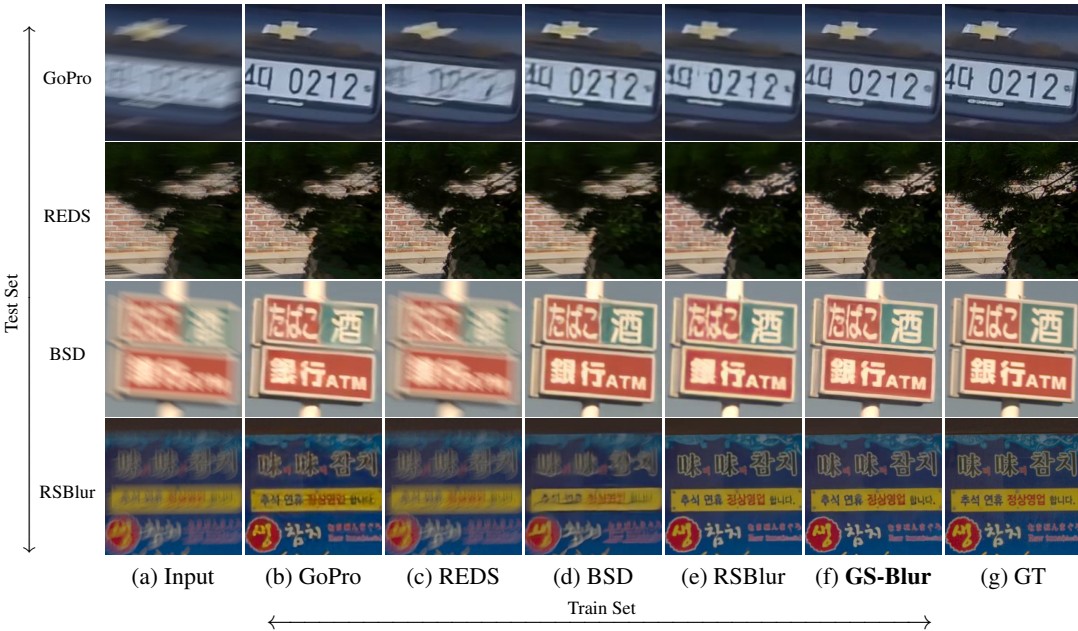

Figure 8: **Qualitative comparison of cross-validation using UFormer [40].**

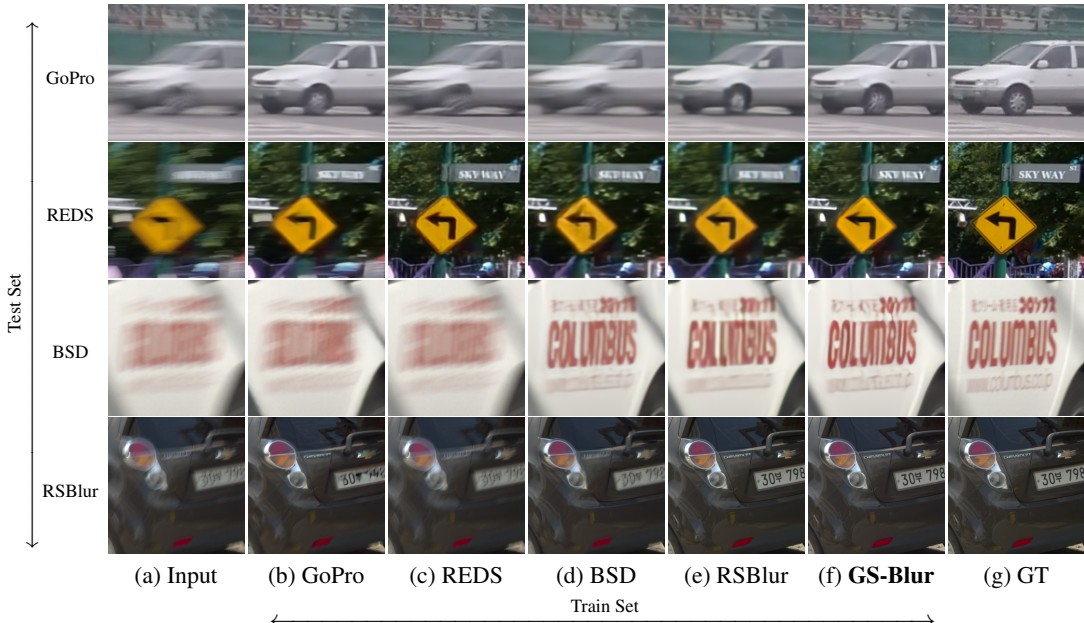

Figure 9: **Qualitative comparison of cross-validation using NAFNet [3].**

## B Selected Classes from MVImgNet

We selected 26 classes from MVImgNet [43] that contain rich 3D information and textures when generating GS-Blur dataset. The IDs and names of the selected classes are as follows: {6: table, 8: sofa, 14: flowerpot, 17: mug, 26: pot, 28: guitar, 29: bookshelf, 36: chair, 37: car, 38: cap, 39: can, 41: cabinet, 44: bicycle, 45: bench, 46: bed, 52: plush toy, 93: coat rack, 112: ladder, 137: rockery, 152: strings, 155: scarf, 156: shoe, 158: pants, 159: clothing}

## C Additional visualization of GS-Blur

In Figure 10 we provide a comprehensive visualization of all the classes we generated in the GS-Blur dataset. This figure highlights the diversity and richness of the dataset across various scenes and blur patterns.

## D Negative societal impact

While advanced deblurring algorithms facilitate easy image enhancement for the general public, their use also raises concerns about potentially malicious applications, particularly regarding privacy issues. Blurring is commonly employed to protect personal information, such as faces and personal IDs. To address potential misuse, image forensic algorithms can be employed, which aim to authenticate images. Many of these algorithms focus on training classifiers to differentiate between images captured in the real world and those processed by deep learning models.

## E License of the used assets

- 3D Gaussian-splatting is a publicly available dataset released under CC BY MIT license.
- MVImgNet dataset is a publicly available dataset released under CC BY 4.0 license.
- GoPro dataset is a publicly available dataset released under CC BY 4.0 license.
- REDS dataset is a publicly available dataset released under CC BY 4.0 license.
- BSD dataset is a publicly available dataset released under CC BY MIT license.
- RSBlur dataset is a publicly available dataset released under CC BY MIT license.

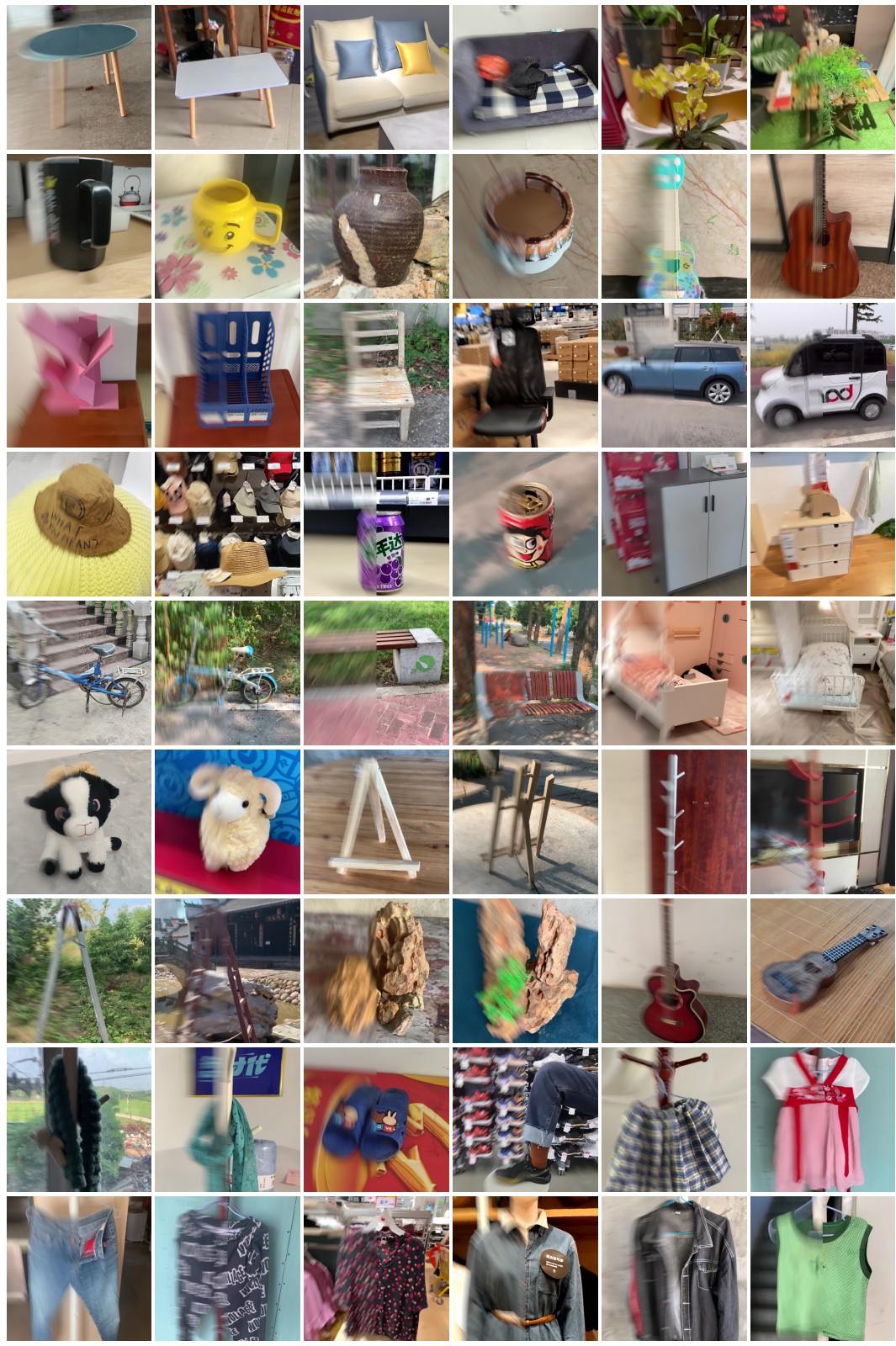

Figure 10: **Examples of the proposed GS-Blur dataset.** The left half of the frames displays synthetically generated blur, while the right half exhibits sharpness.

