# OpenReview forum: "GS-Blur: A 3D Scene-Based Dataset for Realistic Image Deblurring"
_NeurIPS.cc/2024/Datasets_and_Benchmarks_Track — NeurIPS 2024 Track Datasets and Benchmarks Poster_

### Official Review · Reviewer_zVqH · 2024-07-21
**A good paper proposing a new way to simulate deblurring dataset**

**Rating:** 10
**Confidence:** 4
**Correctness:** Yes
**Clarity:** Yes

**Review:**

As the current "forward models" for the simulation of blur mostly depend either on simplistic 2D assumptions or hard-to-capture methods, GS-blur opens up a new path for accurate simulation of the forward blurring model, therefore potentially improving the performance of the deblurring networks. The method is straightforward and clever, and has direct real world impact. One of the best papers I have encountered while reviewing the D&B papers.

**Strengths:**

1. The paper is well written and easy to follow.

2. Clear motivation with direct real-world impact

3. The results are significant, clearly showing the superiority of the method over the other baselines.

**Additional Feedback:**

The method is seemingly obvious in retrospect. But all good ideas are simple in retrospect. Great paper, well done!

**Documentation:**

Yes

**Limitations:**

The limitation is that this method will only be applicable to video captures rather than single images, potentially limiting its applicability.

**Opportunities For Improvement:**

Obviously aiming for better 3D reconstruction would lead to improved performance, but I believe this is beyond the scope of the paper for now.

**Relation To Prior Work:**

Yes

**Summary And Contributions:**

The paper proposes GS-blur, a method of synthesizing a realistic blurry image from a clean image, is proposed. The method is based on first reconstructing the 3D scene through GS, and simulating the motion blur through proposing a 3D camera trajectory, which averages the captured frames. Comparing the proposed simulation method against others by training SOTA deblurring networks, it is clearly shown that the proposed forward model is the most robust and high-performing.

---

> ### Author Rebuttal · Authors · 2024-08-16
>
> We sincerely appreciate the reviewer’s thoughtful and encouraging feedback, and we are grateful for the recognition of our work's potential impact on advancing motion deblurring task.
>
> **[Better 3D reconstruction model]** We also believe that improved 3D reconstruction [1, 2], would lead to a more accurate dataset reconstruction of our GS-Blur, since GS-Blur leverages multi-resolution renderings.
> We plan to release code that synthesizes blurry images from 3DGS along with our GS-Blur dataset, in order to increase accessibility and enable the application of the proposed approach with better 3D reconstruction methods.
>
> **[Apply GS-Blur to single image capturing]** We believe that recent advancements in 3D reconstruction or generation from a single image [3, 4] could enhance our method, making it applicable not only to video capture but also to single images in the future.
> In this case, it seems necessary to more delicately consider the occlusion and disocclusion at the object boundaries caused by camera motion.
> We are grateful for the reviewer's insights, which have helped us consider these possibilities.
>
> Reference:
> [1] Zehao Yu, Anpei Chen, Binbin Huang, Torsten Sattler and Andreas Geiger. Mip-splatting: Alias-free 3d gaussian splatting. In CVPR 2024.
> [2] Shakiba Kheradmand, Daniel Rebain, Gopal Sharma, Weiwei Sun, Yahg-Che Tseng, Hossam Isack, Abhishek Kar, Andrea Tagliasacchi and Kwang Moo Yi. 3D Gaussian Splatting as Markov Chain Monte Carlo. In arXiv preprint arXiv:2404.09591.
> [3] Stanislaw Szymanowicz, Christian Rupprecht and Andrea Vedaldi. Splatter Image: Ultra-Fast Single-View
> 3D Reconstruction. In CVPR 2024.
> [4] Zi-Xin Zou, Zhipeng Yu, Yuan-Chen Guo, Yangguang Li, Ding Liang, Yan-Pei Cao and Song-Hai Zhang. Triplane Meets Gaussian Splatting: Fast and Generalizable Single-View 3D Reconstruction with Transformers. In CVPR 2024.

---

### Official Review · Reviewer_h9V9 · 2024-07-22
**Limitations of the GS-Blur dataset**

**Rating:** 6
**Confidence:** 4
**Correctness:** Yes
**Clarity:** Yes

**Review:**

The motivation and ideas are clear. This paper proposes an interesting way to generate a deblurring dataset. However, I have a few concerns.

1. As mentioned in lines 150-154, the sharp images are rendered from a fixed camera position. In the following training step, the sharp images will be taken as the ground truth. However, I am wondering if the sharp images are truly "ground truth." From my understanding, the sharp images rendered by 3DGS may not be accurate. If so, how do the authors avoid inaccuracies?

2. Could the authors provide more explanation about the comparison of different datasets? Why does the model trained on the proposed GS-Blur dataset achieve better performance than that trained on real blur datasets? Is the performance improvement due to the size of the dataset?

3. From my understanding, the training of 3DGS is not expensive. Thus, I am wondering if the authors conducted ablation studies on the scale of the GS-Blur dataset.

**Strengths:**

1. The motivation and ideas are clear. Using 3DGS to generate the deblurring dataset is an interesting approach.

2. The experimental results demonstrate the effectiveness of the proposed dataset.

**Additional Feedback:**

No

**Documentation:**

Yes

**Ethics:**

No.

**Limitations:**

Yes

**Opportunities For Improvement:**

My suggestion is that the author should provide in-depth discussions about the scale of the training data since the training of 3DGS is not expensive.

**Relation To Prior Work:**

Yes

**Summary And Contributions:**

This paper proposes GS-Blur, a new dataset of synthesized realistic blurry images created using a novel approach. The authors first reconstruct 3D scenes from multi-view images using 3D Gaussian Splatting (3DGS), then render blurry images by moving the camera view along randomly generated motion trajectories. By adopting various camera trajectories in reconstructing GS-Blur, this dataset can feature realistic and diverse types of blur, offering a large-scale dataset that generalizes well to real-world blur.

---

> ### Author Rebuttal · Authors · 2024-08-16
>
> Thank you for the reviewer's valuable comments, which have provided us with insightful guidance to clarify our work.
>
> **[Using rendered images as ground truths]**
> We acknowledge the reviewer's concerns about using sharp images rendered by 3DGS as ground truths. If the 3D reconstruction is well-converged, the sharp image can be considered as the ground truth since we synthesize the blurry pair by aggregating these renderings. However, inaccuracies can occur during the optimization of 3DGS, leading to floating artifacts or unrealistic geometry in novel views. To ensure quality, we measure PSNR between ground truth images and renderings. If any view exhibits a drop in PSNR of more than 3dB from the mean, the scene is classified as failed for 3D reconstruction. For details, please refer to our response to Reviewer **EMvE**.
>
> **[More explanation about the performance improvement]**
> There are two primary reasons why the model trained on our GS-Blur dataset outperforms those trained on real-blur datasets. First, as mentioned in lines 36-38 of our main manuscript, previous real-blur datasets required specialized and rigid camera systems, which limited the variety of camera models used to capture the images. Since different camera models employ different ISP processes, this limitation reduced the diversity in the appearance of the images. In contrast, our GS-Blur dataset is reconstructed using MVImgNet, which is sourced from a wide range of devices through crowd-sourcing in real-world scenarios. This results in greater diversity and improved generalizability in cross-validation.
>
> Second, as mentioned in lines 41-42 and shown in Figure 2 of our main manuscript, previous real-blur datasets heavily relied on human-operated systems, using dual-camera setups with tripods and external trigger equipment. This approach made it difficult to capture the complex diversity of potential motion trajectories. In contrast, our method can accommodate any type of motion trajectory that can be represented by a Bezier curve, leading to superior performance.
>
> **[More ablation about the scale of GS-Blur dataset]**
> While our method allows for the generation of deblurring dataset pairs at a low cost, _Table 1_ further explores the impact of different dataset scales. Here, we use NAFNet as the baseline model and match the image scales to those of the previous datasets, GOPRO, RSBlur, BSD, REDS. As shown, the performance improves gradually as the number of images in the dataset increases. Additionally, when compared to Table 2 in our main manuscript, our GS-Blur consistently shows superior generalizability even when scaled to match the sizes of the aforementioned datasets. This indicates that the advantages of our GS-Blur dataset extend its cost-effectiveness in dataset generation; it is also effective in modeling diverse motion trajectories.
>
> _Table 1. Ablation study on deblurring performance across different dataset scales matched to previous datasets._
> |          |          | GoPro      |         | REDS       |         | BSD        |         | RSBlur     |         |
> |----------|----------|------------|---------|------------|---------|------------|---------|------------|---------|
> | # Images | Matched to | PSNR      | SSIM    | PSNR       | SSIM    | PSNR       | SSIM    | PSNR       | SSIM    |
> | 2103     | GoPro      | 30.55     | 0.941   | 29.78      | 0.913   | 30.73      | 0.927   | 31.80      | 0.860   |
> | 8878     | RSBlur     | 31.15     | 0.946   | 30.17      | 0.919   | 31.11      | 0.931   | 32.13      | 0.865   |
> | 18000    | BSD        | 31.32     | 0.947   | 30.25      | 0.921   | 31.25      | 0.932   | 32.24      | 0.866   |
> | 24000    | REDS       | 31.41     | 0.948   | 30.35      | 0.922   | 31.23      | 0.933   | 32.22      | 0.866   |
> | 752335   | GS-Blur    | 31.49     | 0.949   | 30.54      | 0.924   | 31.37      | 0.941   | 32.30      | 0.868   |

---

### Official Review · Reviewer_EMvE · 2024-07-25
**A valuable deblurring dataset created using 3dgs**

**Rating:** 7
**Confidence:** 3
**Correctness:** Yes.
**Clarity:** Yes.

**Review:**

The authors present a novel dataset, GS-Blur, which synthesizes blurry images using 3D Gaussian Splatting (3DGS). Several deblurring methods are benchmarked on GS-Blur and other datasets. Qualitative comparison of cross-validation shows the data quality of GS-Blur. The authors provide a detailed explanation of the dataset construction process. They also demonstrate the quality of the dataset through ablation studies. Therefore, this article is qualified to be accepted.

**Strengths:**

The scale of GS-Blur is larger than previous deblurring datasets. Qualitative comparison of cross-validation shows that the data quality of GS-Blur is better than other datasets.

**Additional Feedback:**

Nothing.

**Documentation:**

The authors will publicly release the dataset, but haven't provide the dataset link by now.

**Ethics:**

No.

**Limitations:**

The authors addressed the limitations at the end of the article.

**Opportunities For Improvement:**

Floating points may also cause blurring effect while carrying out novel view synthsis using 3dgs. I wonder if it will influence the quality of GS-Blur.

**Relation To Prior Work:**

Yes.

**Summary And Contributions:**

- The authors present a novel dataset, GS-Blur, which synthesizes blurry images using 3D Gaussian Splatting (3DGS).
- Several deblurring methods are benchmarked on GS-Blur and other datasets.

---

> ### Author Rebuttal · Authors · 2024-08-16
>
> We appreciate the reviewer's concern regarding the potential blurring effects caused by floating points during novel view synthesis using 3DGS.
>
> **[How to minimize the impact of floating points artifact]** To address this, we have measured the PSNR (Peak Signal-to-Noise Ratio) between the ground truth images and the renderings for each 3D reconstructed scene. In cases where any view exhibited a drop in PSNR of more than 3dB from the mean PSNR, we classified the entire scene as a failed scene. This approach ensures that only high-quality scenes contribute to the final results, minimizing the impact of potential floating point-induced blurring on GS-Blur dataset.
>
> Using this method, we clustered 1622 scenes (32%) that did not meet the PSNR threshold as failed.
> Training NAFNet with these failed scenes produced the results shown in Table 1.
> When training was conducted using only the scenes that failed PSNR thresholding (the first row), there was a noticeable drop in deblurring performance.
> This result shows that scenes with inaccurate 3D reconstruction (e.g., floating points) negatively impact the training of deblurring networks.
> Additionally, when trained on our dataset (the third row), which filters out scenes that do not meet the PSNR threshold, the network consistently outperforms the one trained without filtering (the second row), even though fewer scenes were used. This indicates that our filtering is effective. We will include this content in our final version of the paper.
>
> _Table 1. Deblurring performance comparison when training NAFNet by PSNR thresholding._
> |         |         | GoPro |       | REDS |       | BSD  |       | RSBlur |       |
> |----------|---------|-------|-------|------|-------|------|-------|--------|-------|
> |  PSNR Thresholding | # Scenes | PSNR  | SSIM  | PSNR | SSIM  | PSNR | SSIM  | PSNR   | SSIM  |
> | Failed     | 1622    | 31.01 | 0.945 | 30.26| 0.919 | 31.36| 0.934 | 32.05  | 0.865 |
> | Passed+Failed     | 5030    | 31.46 | 0.948 | 30.52| 0.923 | __31.42__| 0.935 | 32.13  | 0.866 |
> | Passed (GS-Blur)     | 3408    | __31.49__ | __0.949__ | __30.54__| __0.924__ | 31.37| __0.941__ | __32.30__  | __0.868__ |

---

### Author Rebuttal · Authors · 2024-08-16

We sincerely appreciate the reviewers for their valuable time and effort in evaluating our work. We are encouraged by their acknowledge of the novelty and efficacy of our proposed GS-Blur. We do our best to address all the concerns raised, and we hope our responses adequately address the reviewers' feedback.

---

### Decision · Program_Chairs · 2024-09-26

**Decision:**

Accept (Poster)

**Comment:**

This paper presents a novel dataset, GS-Blur, which synthesizes blurry images using 3D Gaussian Splatting (3DGS). After repeated communication between the author and the reviewers, the relevant doubts and uncertainties were resolved. Ultimately, based on various factors, including innovation and rigor, I have determined that this paper can be accepted.